# Effects of Salinity Levels of Drinking Water on Water Intake and Loss, Feed Utilization, Body Weight, Thermoregulatory Traits, and Blood Constituents in Growing and Mature Blackhead Ogaden Sheep and Somali Goats

**DOI:** 10.3390/ani14111565

**Published:** 2024-05-25

**Authors:** Hirut Yirga, Mengistu Urge, Arthur Louis Goetsch, Adugna Tolera, Ryszard Puchala, Amlan Kumar Patra

**Affiliations:** 1School of Animal and Range Sciences, Haramaya University, Dire Dawa P.O. Box 138, Ethiopia; hirut.tejeji@langston.edu (H.Y.); urgeletta@yahoo.com (M.U.); 2American Institute for Goat Research, Langston University, Langston, OK 73050, USA; arthur.goetsch@langston.edu; 3School of Animal and Range Sciences, Hawassa University, Hawassa P.O. Box 05, Ethiopia; adugnatolera3@gmail.com; 4Military Institute of Hygiene and Epidemiology, Kozielska 4, 01-163 Warsaw, Poland

**Keywords:** goats, performance, salinity level, sheep

## Abstract

**Simple Summary:**

Livestock in different parts of the world consume drinking water containing variable levels of total dissolved solids (TDSs) that may influence their performance and health. Growing and mature Blackhead Ogaden sheep and Somali goats were used to evaluate the effect of salinity levels (up to 17 g/L of TDSs) of drinking water on water intake and loss, nutrient utilization, thermoregulation, and blood characteristics. Drinking water, total water intake, urine excretion, and total water loss were higher in both young and mature sheep and goats, while apparent dry matter digestibility decreased for high levels of TDSs compared with fresh water. There were no adverse effects of high salinity levels tested on feed intake, body weight change, different blood variables, and thermoregulatory mechanisms, reflecting considerable tolerance of relatively high levels of salinity in drinking water by these goat and sheep breeds.

**Abstract:**

This experiment was conducted to evaluate the effects of drinking water salinity levels on water intake and loss, feed intake and digestion, body weight (BW), thermoregulation, and blood characteristics on growing and mature (18.8 ± 0.39 and 21.8 ± 0.40 kg BW, and 0.6–1 and 1.5–2 years of age, respectively) Blackhead Ogaden sheep and Somali goats. The animals were assigned to a 4 (water salinity) × 2 (sheep and goat species) × 2 (growing and mature animals) factorial arrangement for the 60-day experimental period and 10-day digestibility determination. Water treatments were fresh water (FRW) and low (SW-L), moderate (SW-M), and high (SW-H) levels of salinity (i.e., the addition of NaCl to obtain 10, 13.5, and 17 g of total dissolved salts (TDSs)/L, respectively). The salinity of drinking water did not affect feed intake, BW, thermoregulatory traits (respiration rate, rectal temperature, and heart rate), or blood parameters (*p* > 0.05); however, drinking water, total water intake, urine excretion, and total water loss increased (*p* < 0.01) while apparent dry matter digestibility decreased quadratically (*p* < 0.01) with increasing water salinity. Analysis of the interaction between water treatment and species showed that PCV (*p* = 0.059) and hemoglobin (*p* = 0.070) levels tended to be higher in sheep than in goats drinking FRW, and AST activities were greater (*p* = 0.036) in goats consuming SW-M than in sheep consuming water with the same salinity level. In conclusion, increasing the salinity level of drinking water by adding NaCl to up to 17 g/L of TDSs had no adverse effect on the water intake, feed intake, BW, and health status of growing and mature Blackhead Ogaden sheep and Somali goats.

## 1. Introduction

Livestock require good quality drinking water to maintain satisfactory production [1,2]. Water of poor quality may affect water palatability, production, and animal health [2,3,4]. Concentrations of total dissolved solids (TDSs) or salinity, pH, hardness, mineral content, temperature, and microbial loads determine water quality [5]. One of the principal factors determining the suitability of drinking water for livestock is salinity [6]. Animals generally drink water with varying concentrations of TDSs depending on water availability [2,3,7]. However, water salinity is projected to increase in many parts of the world due to climate change, and progressive salinization is a serious global issue [5,6,8]. Drinking water with TDS tolerance threshold levels <10 g/L may increase feed and water intake by animals [4]; however, water with salinity beyond this level is not usually recommended for livestock consumption [3,6]. Excessive salt intake interferes with physiological homeostasis and routine biological processes and may affect the health and productive performance of animals [9]. Changes in water intake and loss, feed intake, body weight (BW), thermoregulatory traits (such as rectal temperature (RT), pulse rate (PR), and respiration rate (RR)), and blood constituents (hematological and biochemical profiles) are common physiological parameters used to evaluate the production efficiency and health status of animals. Consumption of too much salt has been reported to reduce feed intake and growth, increase water requirement, and affect thermoregulation and blood parameters [9,10].

Though animals can consume saline water, tolerance to the salt content in drinking water varies with species, breed, age, physiological condition, and salt content of the water and diet [7,11,12]. The tolerance of goats to water with high salinity has been noted to be greater than that of sheep [3] but lower than those of other livestock species [4,6]. Different breeds of young and mature sheep and goats were tolerant to brackish groundwater with TDS levels up to 10 g/L containing different types of dissolved salts, and there were no interactions between breed and water type [7,12]. Breeds of ruminants that are well adapted to desert environments demonstrate a greater ability to ameliorate the stressful effects induced by drinking salty water [9,13]. Younger classes of livestock are less tolerant to water with high salinity compared with mature dry stock [6].

Considering the predicted increase in water salinity in the future and the variations in salinity tolerance among animals [7], it is necessary to identify the detrimental effects of water salinity on the performance and health of animals and to determine the upper safe level that can be tolerated by animals. This would enable livestock producers and policymakers to assess the ability of indigenous animals to survive and reproduce when subjected to saline water. However, the information available on the effect of drinking water salinity on the performance of animals varies across experiments based on the breeds used, the geographical area, and the methods employed, and the results were inconclusive for practical use [5]. Blackhead Ogaden sheep and Somali goats are well adapted to the arid and semi-arid areas of Ethiopia and Somalia, and they are among the promising breeds of small ruminants due to their better adaptability to the harsh climatic conditions [14]. The tolerability of these animals to water salinity might be greater due to their adaptability to arid conditions; however, their water salinity tolerance levels have not been well-characterized. Hence, the objective of this experiment was to evaluate the impact of salinity levels of drinking water on the performance (water intake, feed intake, digestion, and BW) and health (thermoregulatory traits and blood constituents) of growing and mature Blackhead Ogaden sheep and Somali goats.

## 2. Materials and Methods

### 2.1. Animals, Management, and Water Treatment

The experiment was conducted at Haramaya University (9°25′ N latitude and 42°2′ E longitude) in eastern Ethiopia. A total of 80 animals from 2 species (Blackhead Ogaden sheep and Somali goats) and 2 age groups (growing and mature) were purchased from local markets and used for this experiment. Age was estimated based on dentition and information obtained from the owners. The age of growing animals was estimated at 0.6–1.0 years, while that of mature animals was estimated at 1.5–2 years. The animals were acclimated to the research area for 4 weeks during which time they grazed on pasture on the university campus and were quarantined so that their health conditions could be observed. The animals were vaccinated against the common infectious diseases (pasteurellosis and anthrax) prevalent in the area, dewormed against internal parasites, and sprayed with acaricides against external parasites. At the end of the quarantine period, each animal was weighed and placed in an individual pen (134.8 cm × 78 cm) equipped with a feeding trough and watering bucket. They were acclimated to the experimental environment, pen, diet, and water for 3 weeks. This was followed by a 60-day experimental period and 10 days of determining digestibility. The water treatments were introduced gradually during the 3rd week of adaptation. When the experiment started, the average body weights (BWs) of growing (GRO-S) and mature (MAT-S) Blackhead Ogaden sheep and growing (GRO-G) and mature (MAT-G) Somali goats were 18.3 ± 0.54, 22.4 ± 0.52, 18.2 ± 0.51, and 22.0 ± 0.54 kg, respectively.

The experiment was a completely randomized block design with a 4 × 2 × 2 factorial arrangement of treatments, with four water treatments, two animal species, and two age groups (Table 1). There were 10 animals in each age group for each species (*n* = 10). Animals in each age and species group were blocked into five sets, with four animals in each set according to their initial BW. The four drinking water treatments were then randomly assigned to animals in a set, with each treatment group containing five animals. Water treatments were fresh potable tap water from the university (FRW) as a control, and saline water with three levels of salinity (added NaCl). The three levels were low (SW-L), moderate (SW-M), and high (SW-H), containing 10, 13.5, and 17 g of TDSs/L, respectively. The levels were based on a maximum upper limit of 10 g of TDSs/L for livestock consumption [4] (SW-L), while SW-M and SW-H were set to increase the TDSs in SW-L by 35 and 70%, respectively. Batches of the water treatments were prepared weekly and stored in clean covered cans.

An amount of water above that being consumed by each animal was placed in a clean bucket twice a day for approximately 30-min periods to allow animals a total of 1 h per day for consumption. The feeds offered were a basal diet of grass hay with 9.22 ± 0.23% ash and 9.19 ± 1.18% crude protein (CP), a supplement of concentrate mixture made up of wheat bran (60%), and noug seed cake (40%) with 23.4 ± 1.18% CP aimed at providing nutrient requirements for maintenance and growth [15]. Grass hay was offered at approximately 120% of the preceding few days for ad libitum consumption, while the supplement was offered at 1.5% of BW per day in equal amounts twice a day, and the amount of feed offered was adjusted based on BWs measured at 10-day intervals. Animals had access to trace-mineralized salt blocks.

### 2.2. Measurements

#### 2.2.1. Water Intake, Feed Intake, and Body Weight

Water offered and refused at both times were weighed and recorded to determine daily drinking water intake by an animal. Water was sampled every time water treatments were prepared. The feed refused by each animal was recorded daily and pooled per treatment. The daily feed intake of an animal was determined based on the weight difference between the offered and refused feed. Samples of offered feeds were taken when new batches of hay and concentrates were used. The combination of water intake from drinking and feeds was considered the total water intake. Animals were weighed to the nearest 0.2 kg using a digital scale at the start of the experiment (initial BW) and every 10 days thereafter before feed was provided. The change in the body weight of each animal was calculated as the difference between final and initial body weights. The average daily gain (ADG) by each animal was determined based on the change in BW and the number of experimental days.

#### 2.2.2. Thermoregulatory Traits

Rectal temperature (RT), pulse rate (PR), and respiration rate (RR) were measured every 10 days at 07:00 h. The rectal temperature of each animal was measured by inserting a digital thermometer (TM01, Cotronic Technology Ltd., Shenzhen, China) carefully into the rectum; recordings were taken to the nearest 0.1 °C once a stable automated reading was obtained. The thermometer was cleaned with an alcohol wipe and lubricated with petroleum jelly before use on each animal. Each animal’s RR was measured visually by counting the flank movements over one minute of regular breathing (time was measured with the aid of a stopwatch) with the animal standing quietly. The PR was taken by counting the number of beats on the artery below and slightly inside the jaw by lightly pressing on the skin with fingers and monitoring for 1 min (time was measured using a stopwatch).

#### 2.2.3. Metabolism Trial

At the end of the 60-day feeding study, animals were fitted with fecal collection bags and placed in individual metabolic cages. Animals were allowed to adapt to carrying the fecal bags and the metabolic cages for 3 days. Feces were collected into the fecal collection bags for 7 days and urine collected in buckets was weighed in the morning before feeding. A total of 10% of the daily feces output of each animal was taken and bulked across the experimental days and kept in a deep freezer at −20 °C. At the end of the collection period, samples from each animal were pooled and 10% was sub-sampled for analysis. Total water loss was determined by summing the water excreted via urine and feces. During this period, daily water intake, feed intake, and body weights were also recorded. Samples of drinking water, feed offered, and feed refused were also bulked over the experimental period and sub-samples were taken for chemical analysis. The apparent digestibility of DM was calculated as the difference between nutrient intake and nutrients recovered in feces, expressed as a proportion of nutrient intake.

#### 2.2.4. Blood Constituents

At the end of the metabolism trial, jugular blood samples were collected from each animal and placed in two 10 mL vacutainer tubes, one without an anticoagulant and one containing EDTA as an anticoagulant. The blood samples in the plain tubes were used for packed cell volume (PCV) and hemoglobin determination. The samples in tubes with EDTA were centrifuged and plasma was separated and transferred into clean plastic vials, which were immediately placed in a refrigerator at −20 °C for use in determining the concentrations of biochemical indices, including total protein, albumin, glucose, total cholesterol, triglycerides, blood urea nitrogen (BUN), creatinine, alkaline phosphatase (ALP), alanine aminotransferase (ALT), and aspartate aminotransferase (AST).

#### 2.2.5. Laboratory Analyses

The water samples were analyzed for electrical conductivity (EC) at Haramaya University’s chemistry laboratory as per the APHA [16] standard test guidelines [17]. The TDS level in the water was calculated based on the EC value using the formula: TDS = EC × 0.64, as per Al Dahaan et al. [18]. Samples of feeds offered and refused and feces were analyzed for dry matter (DM), ash, and Kjeldahl nitrogen (N) according to the AOAC procedures [19]. Crude protein (CP) contents in feed and feces were obtained by multiplying N × 6.25. The PCV and hemoglobin in the blood [20] were measured using an automated hematology analyzer (Sysmex KX-21, Kobe, Japan). Serum was analyzed for total protein (Biuret method), albumin (Bromocresol green method), urea (enzymatic GLDH method), creatinine (Jaffe’ kinetic method), glucose (enzymatic GOD-PAP method), triglyceride (GPO-PAP method with lipid clearing factor), and cholesterol (CHOD-PAP method with lipid clearing factor) concentrations, and ALT, AST [21], and ALP [22] activities were colorimetrically determined using a semi-automatic biochemistry analyzer (HumaLyzer 3000, Wiesbaden, Germany) using the reagents prepared for each test.

### 2.3. Statistical Analyses

General linear model procedures for analyzing data with one observation per animal (blood variable) and mixed effects model [23] procedures for analyzing variables measured at different time points were used. The models contained water treatment, animal species, age, and their interactions based on the equation below:Y_ijkl_ = μ + T_i_ + S_j_ + T × S_ij_ + A_k_ + T × A_ik_ + S × A_jk_ + T × S × A_ijk_ + AN_l_(_ijk_) + e_ijkl_,
where Y_ijkl_ = dependent variable, μ = overall mean, T_i_ = effect of ith water treatment, S_j_ = effect of jth species, T × S_ij_ = effect of the interaction between water treatment and species, A_k_ = effect of age, T × A_ik_ = effect of interaction between water treatment and age, S × A_jk_ = effect of the interaction between species and age, T × S × A_ijk_ = effect of the interaction between water treatment, species, and age, AN_l_(_ijk_) = the random effect of an animal in the water treatment, species, and age group, and e_ijkl_ = residual error.

Species and age group means were separated using the least significant difference test, with the protected F-test (*p* < 0.05). Interaction means are listed when an interaction is significant (*p* < 0.05), and main effect means are included when the factor is not involved in an interaction, regardless of the significance of the main effect. Orthogonal contrasts were also performed for the addition of salt to drinking water (FRW vs. mean of SW-L, SW-M, and SW-H) and linear and quadratic effects of salinity levels of SW-L, SW-M, and SW-H. These contrasts are stated when they are significant (*p* < 0.05).

## 3. Results and Discussion

### 3.1. Water Intake and Losses

TDS levels in the water with added salt were slightly greater than planned (TDS levels of 540, 10,620, 13,650, and 18,160 mg/L in FRW, SW-L, SW-M, and SW-H, respectively). This may have been due to the presence of other ions such as Ca, K, and Mg, above the Na and Cl, that contribute to TDS levels [4]. Levels of Na and Cl in water with added salt were above the maximum levels recommended for livestock drinking, i.e., the TDS levels were above 10 g/L [4] and 14 g/L [3], the recommended maximum levels in livestock drinking water.

There was no significant interaction between water intake and loss (Table 2 and Table 3). Greater intakes of drinking water, feed water, and total water (*p* < 0.001), but lower intake relative to DM intake (*p* = 0.042), were observed in mature versus growing animals. The observed variation between age groups concurs with the NRC report [1]. High water intake by mature versus growing animals might be because of increased feed intake and water requirement for proper digestion due to the larger body size [24]. Lower water intake relative to DM intake in mature animals compared with growing animals might be due to higher feed intake per kg of metabolic body size (76.3 vs. 69.7 g/kg BW^0.75^) and the associated feed water intake and metabolic water production.

Fecal and total water losses (*p* = 0.026 and 0.035, respectively) by mature animals increased by 14.7% and 15.9% compared with water losses by growing animals. Urinary, fecal, and total water losses (*p* = 0.038, 0.018, and 0.007, respectively) were higher in sheep than in goats (24.6, 15.7, and 20.1%, respectively).

Water treatment influenced drinking water intake (*p* < 0.001), with more SW-M and SW-H being consumed compared with FRW (Table 2 and Table 3). Total SW-M and SW-H water intakes (mL/day, mL/g DMI, and g/kg BW^0.75^) were also greater (*p* < 0.001) than FRW intake; however, FRW and SW-L intakes were similar (*p* > 0.05). The drinking water intake (mL/day) and the total water intake (as mL/day, mL/g DMI, and g/kg BW^0.75^) increased by 17.7, 16.7, 14.2, and 17.2%, respectively, for water with added salt than for FRW. In agreement with this study, many studies have indicated that high water salinity increases water intake in sheep [7,10,13] and goats [7,25,26]. It also increases urine excretion [13,27]; however, other researchers [27,28] have reported a lack of its effect on water intake. In studies of three breeds of young and mature sheep and goats, drinking water intake increased due to increased salinity levels, although there was no interaction between animal type and water treatment, indicating that sheep and goats responded similarly [7,12].

Higher levels of salinity in drinking water increase salt concentrations in the body; thus, animals attempt to regulate osmotic pressure and the concentrations of salts in body fluids to within physiological limits [11]. Hence, the purpose of increasing water consumption would be to facilitate increased urine output to remove the salt load because there is little or no capacity to store excess electrolytes [7,9]. Nevertheless, Zoidis and Hadjigeorgiou [29] noted that water intake by goats increased when water contained up to 10 g of TDSs/L, but reduced when it contained 20 g/L of TDSs. Similarly, Attia-Ismail et al. [30] reported increased water intake by sheep and goats in response to drinking water containing 8.15 g of TDSs/L, but reduced at 12.3 g/L of TDSs. In contrast to this finding, other studies have indicated that higher salinity levels decrease water consumption by goats [11] and calves [8].

Urine and total water loss (*p* < 0.001) were greater for water with added salt than for freshwater (64.4% and 39.9%, respectively, for water with added salt and for FRW) and showed a quadratic response (*p* = 0.013 and 0.040, respectively), with the highest loss occurring following SW-M intake. This agrees with studies in sheep [13] and goats [27,29], which found higher urine excretion due to higher levels of water salinity. There was no effect of water salinity on fecal water loss (*p* > 0.05). Similar to this finding, a nonsignificant effect of water salinity on fecal water loss has been noted previously [8,25,28]. The increase in water loss due to drinking water with added salt is similar to the increase in water intake. Animals excrete more urine and increase their glomerular filtration rates to remove the salt load from their bodies.

### 3.2. Feed Intake, Digestibility, and Body Weight Changes

There were no interaction effects for feed intake, growth rate, and digestibility (Table 4 and Table 5). There was no difference between water salinity and DM intake (*p =* 0.91), but there were differences between age groups and species (*p <* 0.05) and DM intake in g/kg BW^0.75^ or percent of BW during the whole experimental period (Table 4 and Table 5). Total DM intake in g/day tended to be greater (*p* = 0.088) for sheep than for goats. Final BW and ADG were not affected (*p* > 0.05) by water salinity or species. In the age groups, feed intake and digested DM were greater in mature than in growing animals (*p* < 0.001).

During the digestibility trial, intakes of total feed DM and OM also tended to be greater (*p* = 0.080) in sheep than in goats. In support of this, studies have found greater DM intake in sheep than in goats [30,31]. Though nutrient digestibility was similar between young and mature animals, DM and OM digestibility tended to be higher (*p* = 0.093 and 0.098, respectively) in goats than in sheep. The decreasing digestibility in sheep might be due to shorter ruminal digesta retention times because of greater DM intake relative to BW or BW^0.75^ in sheep than in goats.

The salinity of drinking water did not affect feed and digested nutrient intake (*p* > 0.05). Orthogonal contrast analysis indicated that nutrient digestion was lower for salt-added water treatments than for FRW (*p* < 0.01). Within salt-added water treatments, DM and OM digestibility increased linearly (*p* < 0.01) with increasing salinity levels. The absence of an effect of water salinity on feed intake agrees with other studies [25,27,28]. Conversely, in some cases, the salinity of drinking water influenced feed intake [10,11,26], while in other instances, digestibility was not affected [8,13,30]. Feed intake decreased in sheep and goats consuming drinking water containing TDS levels below or closer to the values used in this study (9.5–17 g/L) [10,32]. In buffalo calves, feed intake reduced even at a TDS level of 8.8 g/L [8]. Tsukahara et al. [27] reported that OM digestibility in goats decreased due to drinking brackish water (6.9 g of TDSs/L). In growing and mature Boer goats and mature Katahdin sheep, OM, CP, and NDF digestibility was lower in growing Boer goats consuming brackish water compared with those consuming fresh water [33]. Yousfi and Ben Salem [10] reported increased apparent digestibility of CP in sheep drinking saline water (15 g NaCl/L).

Feed digestibility in the rumen depends on ruminal microbial activities and particulate passage rate (or digesta retention time). Reduction in digestibility following the consumption of saline water compared with freshwater can be linked to high salt intake from increased saline water consumption since consumption of excessively high concentrations of salt can affect the functioning of rumen microbes, thus reducing the fermentation of fiber [9,27,34]. The reasons for the increased digestion of SW-M and SW-H compared with SW-L are unclear, but it may be related to the slower passage rate of digesta particles and longer retention of digesta in the rumen, thus increasing the ruminal degradation of nutrients [35].

This study revealed similarities in ADG between animal species, age groups, and water treatments (*p* > 0.05; Table 5). It agrees with other studies [25,26,28,29] that did not find any effect of drinking water salinity on ADG by animals; however, others reported decreased ADG by animals due to drinking saline water [11,13]. The absence of water treatment effect on feed and digested nutrient intake can justify the similarities in the BWs of animals since the changes in body weights of animals are presumably the direct consequence of changes in feed consumption [25]. Though the negative effect of water salinity on animal performance has been stated, the absence of an effect on ADG in animals may be more correlated with breeds of sheep and goats, as well as diet type and quality.

### 3.3. Blood Constituents

There were no differences between species and age groups (*p* > 0.05; Table 6 and Table 7) for most blood parameters. This contrasts with the reports by Egbe-Nwiyi et al. [36] and Khanvilkar et al. [31] who found differences between species. Khanvilkar et al. [31] found greater values of hematological variables, BUN, creatinine, and ALP in sheep compared with goats, but Egbe-Nwiyi et al. [36] found lower values of hematological parameters for sheep than for goats. Similarities in hematological values between age groups agree with studies of goats [37] and cattle [38]; however, an increase in hematological variables in sheep and goats with increasing age has been noted by Egbe-Nwiyi et al. [36]. Njidda et al. [39] reported that values of hematological profiles, BUN, and creatinine in sheep increased with age. The levels of glucose and ALT tended to be lower (*p* = 0.067 and 0.091, respectively; Table 6 and Table 7) in growing animals than in mature animals. The reduction in glucose and ALT may be closely associated with nutrient metabolism and insulin activity, as growing animals have higher metabolic activities than mature animals.

Water treatment had no effect on the concentrations of most blood parameters (*p* > 0.05; Table 6 and Table 7). This agrees with studies that indicated an absence of drinking water salinity on levels of HB, PCV, and glucose in Boer and Spanish goats [12,27]; on the levels of glucose, creatinine, BUN, total protein, total cholesterol, triglycerides, and ALT in Baluchi lambs [13]; and on the levels of albumin, total protein, and triglycerides in sheep [7,10,12]. In contrast, other studies reported an effect of water salinity on the levels of glucose, BUN, and creatinine in goats [29].

Interaction between species and water treatment influenced levels of PCV, hemoglobin, and AST (*p* = 0.059, 0.070, and 0.036, respectively; Table 6). PCV and hemoglobin levels tended to be higher in sheep than in goats drinking FRW. Additionally, the levels of AST were greater in goats consuming SW-M than in sheep consuming water with the same salinity level. Reduced hematological levels in goats compared with sheep could be due to lower nutrition levels since goats consumed less feed compared with sheep (675 vs. 724 g/day; Table 5), and higher AST levels in animals consuming SW-M could be due to lower water intake by goats compared with sheep. The concentrations of blood parameters (except PCV, hemoglobin, and AST in goats) were unchanged following saline water consumption, and the values were within normal physiological limits [40]. The lack of an effect of water salinity on the concentrations of creatinine and BUN (indicators of kidney function), and albumin, total protein, and enzymes (indicators of liver function) suggest that water TDS levels of up to 17 g/L did not likely cause pathological alterations in liver and kidney tissues [10,12]. There were higher ALT, AST, ALP, creatinine, and urea concentrations in the sera of Bakri [41], Hararghe-Highland [42], and Barbarine [10] sheep consuming saline drinking water. These findings may indicate a tolerance of the studied sheep and goat breeds for saline water.

### 3.4. Thermoregulatory Traits

Thermoregulatory traits were similar between age groups and water treatments (*p* > 0.05); however, species effects (*p* < 0.001) were observed for rectal temperature (Table 6 and Table 8). Similarities between age groups contrasted with the study by Naik et al. [38], which indicated that the rectal temperatures, respiration rates, and pulse rates of young Punganur cattle were higher than those of adult cattle. The rectal temperatures of sheep were 0.5 °C higher than those of goats (*p* < 0.001). Greater rectal temperatures in sheep compared with goats agree with the results of Khanvilkar et al. [31]. This could be attributed to reduced metabolic heat production in goats compared with sheep because of their lower feed intake (g/kg BW^0.75^). In agreement with this study, there was an absence of water salinity effect on rectal temperature and respiration rate in animals in another study [13]. In contrast, increased pulse rate due to saline water consumption has been reported in Nguni Goats consuming drinking water containing 11 g of TDSs/L [11]. Hararghe-Highland lambs also showed increased rectal temperatures and respiration rates when they drank water containing 2.56 g/L or greater of TDSs [42]. The absence of an effect on thermoregulation traits in the present study indicates that these sheep and goat breeds, which are adapted to tropical environments, were tolerant to drinking water with salinity levels of up to 17 g of TDSs/L.

## 4. Conclusions

There were differences between sheep and goats in terms of variables such as nutrient intake and rectal temperature, with higher values observed for sheep. Water intake and loss, and intakes of feed and digested nutrients were greater in mature animals than in growing animals. Drinking water with TDS concentrations of up to 17 g/L of added salt increased water intake and water loss and reduced nutrient digestibility compared with drinking fresh water, but had no effect on feed intake, ADG, thermoregulation, or most blood traits associated with nutrient metabolism and health. These results suggest that Blackhead Ogaden sheep and Somali goats that are more than 6 months old and are adapted to arid conditions may maintain normal physiology for 60 days, without apparent adverse effects on health, welfare, and growth performance, when consuming saline drinking water with up to 17 g of TDSs/L. The lactation performance of these sheep and goat breeds, which need higher amounts of drinking water, should be evaluated to determine their tolerance levels for saline water.

## Figures and Tables

**Table 1 animals-14-01565-t001:** Experimental design and animal distribution in the study.

Species	Age	Treatment ^1^			
		FRW	SW-L	SW-M	SW-H
Blackhead Ogaden sheep	Growing	10	10	10	10
	Mature	10	10	10	10
Somali goats	Growing	10	10	10	10
	Mature	10	10	10	10

^1^ FRW, SW-L, SW-M, and SW-H = fresh water and saline water with 10, 13.5, and 17 g TDS/L, respectively.

**Table 2 animals-14-01565-t002:** *p* values of the effects of drinking water treatment (WT) with different salinity levels on water intake and loss (mL/day) in growing and mature Blackhead Ogaden sheep and Somali goats.

Item ^1^	Species	Age	WT	Species × Age	Species × WT	Age × WT
Drinking water intake	0.927	<0.001	<0.001	0.733	0.936	0.801
Feed water intake	0.094	<0.001	0.860	0.665	0.970	0.974
Total						
mL/day	0.839	<0.001	<0.001	0.720	0.938	0.808
mL/g DMI	0.107	0.042	<0.001	0.693	0.944	0.941
mL/kg BW^0.75^	0.647	0.944	<0.001	0.783	0.893	0.797
Urinary water loss	0.038	0.169	<0.001	0.634	0.798	0.290
Fecal water loss	0.018	0.026	0.243	0.666	0.512	0.776
Total fecal and urinary water loss	0.007	0.035	<0.001	0.914	0.505	0.322

^1^ BW^0.75^ = metabolic body size; DMI = dry matter intake.

**Table 3 animals-14-01565-t003:** Effects of drinking water treatment with different salinity levels on water intake and loss (mL/day) in growing and mature Blackhead Ogaden sheep and Somali goats.

Parameters ^1^	Species		Age ^2^			Water Treatment ^3^				
	Goat	Sheep	Gro	Mat	SEM ^4^	FRW	SW-L	SW-M	SW-H	SEM
Drinking water intake	1138	1142	1054 ^b^	1227 ^a^	33.4	999 ^c^	1105 ^bc^	1302 ^a^	1155 ^b^	47.3
Feed water intake	79.6	85.3	72.8 ^b^	92.1 ^a^	2.34	83.2	84.4	81.2	80.9	3.30
Total										
mL/day	1218	1228	1127 ^b^	1319 ^a^	34.6	1082 ^c^	1190 ^bc^	1383 ^a^	1236 ^b^	48.9
mL/g DMI	1.73	1.61	1.75 ^a^	1.59 ^b^	0.053	1.45 ^c^	1.56 ^bc^	1.96 ^a^	1.71 ^b^	0.074
mL/kg BW^0.75^	121	119	121	119	3.1	106 ^b^	115 ^ab^	137 ^a^	123 ^a^	4.39
Urinary water loss	251 ^b^	333 ^a^	265	319	22.3	124 ^c^	279 ^b^	430 ^a^	335 ^ab^	31.6
Fecal water loss	311 ^b^	369 ^a^	313 ^b^	367 ^a^	16.9	299	360	359	342	23.9
Total fecal and urinary water loss	561 ^b^	702 ^a^	577 ^b^	686 ^a^	30.7	422 ^b^	639 ^a^	789 ^a^	677 ^a^	43.4

^1^ BW = body weight, DMI = dry matter intake; ^2^ Gro = growing, Mat = mature, SEM = standard error of the mean; ^3^ FRW, SW-L, SW-M, and SW-H = fresh water and saline water with 10, 13.5, and 17 g of TDSs/L, respectively; ^4^ SEM = standard error of the mean for species or age. ^a,b,c^ Means without a common superscript letter in a row within a grouping differ (*p* < 0.05).

**Table 4 animals-14-01565-t004:** *p* values of the effects of drinking water treatment (WT) with different salinity levels on feed intake, digestion, and body weights of growing and mature Blackhead Ogaden sheep and Somali goats.

Item ^1^	Species	Age	WT	Species × Age	Species × WT	Age × WT
Total experimental period						
DM intake (g/day)	0.088	<0.001	0.906	0.656	0.967	0.975
DM intake (% BW)	0.017	0.035	0.678	0.926	0.988	0.955
DM intake (g/kg BW^0.75^)	0.028	<0.001	0.787	0.843	0.989	0.977
Initial body weight (kg)	0.554	<0.001	0.973	0.801	0952	0.888
Final body weight (kg)	0.591	<0.001	0.880	0.815	0.998	0.949
Average daily gain (g/day)	0.985	0.450	0.293	0.358	0.901	0.297
During digestibility trial						
DM intake (g/day)	0.078	<0.001	0.804	0.927	0.766	0.855
DM digestibility (%)	0.093	0.838	0.486	0.473	0.948	0.741
Digestible DM intake (g/day)	0.823	0.003	0.892	0.720	0.707	0.954
OM intake (g/day)	0.080	<0.001	0.801	0.928	0.765	0.859
OM digestibility (%)	0.098	0.753	0.464	0.473	0.926	0.728
Digestible OM intake (g/day)	0.894	0.002	0.875	0.742	0.687	0.950

^1^ DM, BW^0.75^, and OM = dry matter, metabolic body size, and organic matter, respectively.

**Table 5 animals-14-01565-t005:** Effects of drinking water treatments with different salinity levels on feed intake, digestion, and body weights of growing and mature Blackhead Ogaden sheep and Somali goats.

Item ^1^	Species		Age ^2^			Water Treatment ^3^				
	Goat	Sheep	Gro	Mat	SEM ^4^	FRW	SW-L	SW-M	SW-H	SEM
During total study period										
DM intake (g/day)	675	724	617 ^b^	782 ^a^	20.1	705	714	692	687	28.3
DM intake (% BW)	3.36 ^b^	3.53 ^a^	3.38 ^b^	3.52 ^a^	0.047	3.50	3.47	3.39	3.42	0.066
DM intake (g/kg BW^0.75^)	71.1 ^b^	75.0 ^a^	69.7 ^b^	76.3 ^a^	1.22	74.0	73.8	72.0	72.3	1.26
Initial body weight (kg)	20.1	20.4	18.2 ^b^	22.2 ^a^	0.37	20.0	20.3	20.3	20.4	0.52
Final body weight (kg)	20.6	21.0	18.7 ^b^	22.9 ^a^	0.46	21.1	21.0	20.7	20.4	0.65
Average daily gain (g/day)	8.19	8.29	6.22	10.3	3.75	15.5	10.2	5.80	1.46	5.28
During digestibility trial										
DM intake (g/day)	701	764	650 ^b^	815 ^a^	25.1	713	756	741	719	35.5
DM digestibility (%)	81.1	77.6	79.1	79.5	1.45	81.5	77.3	79.3	79.2	2.03
Digestible DM intake (g/day)	599	590	529 ^b^	660 ^a^	29.1	584	580	591	622	41.1
OM intake (g/day)	637	694	591 ^b^	740 ^a^	22.7	648	687	673	653	32.1
OM digestibility (%)	83.0	80.0	81.2	81.8	1.29	83.4	79.5	81.7	81.5	1.81
Digestible OM intake (g/day)	557	552	493 ^b^	616 ^a^	26.3	543	542	553	581	37.2

^1^ DM, BW, and OM = dry matter, body weight, and organic matter, respectively; ^2^ Gro = growing, Mat = mature; ^3^ FRW, SW-L, SW-M, and SW-H = fresh water and saline water with 10, 13.5, and 17 g of TDS/L, respectively. ^4^ SEM = standard error of the mean for species or age. ^a,b^ Means without a common superscript letter in a row within a grouping differ (*p* < 0.05).

**Table 6 animals-14-01565-t006:** *p* values of the effects of drinking water treatment (WT) with different salinity levels on blood constituents and thermoregulatory traits of growing and mature Blackhead Ogaden sheep and Somali goats.

Parameter ^1^	Species	Age	WT	Species × WT	Species × Age	Age × WT
PCV (%)	0.266	0.474	0.723	0.059	0.262	0.377
Hemoglobin (g/dL)	0.304	0.479	0.728	0.070	0.226	0.425
BUN (mg/dL)	0.300	0.482	0.550	0.379	0.458	0.778
Creatinine (mg/dL)	0.872	0.928	0.533	0.963	0.191	0.504
Glucose (mg/dL)	0.507	0.067	0.552	0.401	0.683	0.375
Triglycerides (mg/dL)	0.356	0.864	0.505	0.443	0.746	0.730
Cholesterol (mg/dL)	0.345	0.950	0.872	0.795	0.909	0.766
Total protein (g/dL)	0.751	0.371	0.447	0.225	0.366	0.391
Albumin (g/dL)	0.197	0.302	0.248	0.225	0.171	0.430
ALP (IU/L)	0.353	0.187	0.538	0.732	0.453	0.152
ALT (IU/L)	0.528	0.091	0.947	0.996	0.784	0.915
AST (IU/L)	0.730	0.398	0.616	0.036	0.218	0.685
Rectal temperature (°C)	<0.001	0.377	0.760	0.905	0.509	0.921
Respiration rate (breath/min)	0.787	0.376	0.714	0.768	0.926	0.601
Pulse rate (beats/min)	0.445	0.347	0.295	0.313	0.224	0.870

^1^ PCV, BUN, ALP, ALT, and AST = packed cell volume, blood urea nitrogen, alkaline phosphatase, alanine aminotransferase, and aspartate aminotransferase, respectively.

**Table 7 animals-14-01565-t007:** Effects of drinking water treatment (WT) with different salinity levels on blood constituents of growing and mature Blackhead Ogaden sheep and Somali goats.

Parameter ^1^	Species		Age ^2^			Water Treatment ^3^				
	Goat	Sheep	Gro	Mat	SEM ^4^	FRW	SW-L	SW-M	SW-H	SEM
PCV (%)	30.7	32.2	30.9	31.9	0.95	32.5	30.4	31.8	31.0	1.34
Hemoglobin (g/dL)	10.3	10.7	10.3	10.7	0.32	10.8	10.2	10.6	10.4	0.45
BUN (mg/dL)	23.8	21.5	23.4	21.9	1.54	24.4	20.7	24.0	21.5	2.18
Creatinine (mg/dL)	1.13	1.14	1.13	1.14	0.052	1.20	1.07	1.17	1.09	0.074
Glucose (mg/dL)	56.3	54.9	53.6	57.6	1.50	57.8	53.4	55.6	55.7	2.12
Triglycerides (mg/dL)	44.2	46.8	45.8	45.3	2.01	45.2	49.1	44.2	43.5	2.84
Cholesterol (mg/dL)	62.5	59.4	60.9	61.1	2.28	63.0	62.1	58.7	61.1	3.23
Total protein (g/dL)	6.67	6.60	6.53	6.73	0.158	6.42	6.87	6.49	6.76	0.22
Albumin (g/dL)	2.61	2.52	2.53	2.60	0.051	2.46	2.65	2.53	2.62	0.073
ALP (IU/L)	120	134	117	136	10.0	109	131	137	130	14.2
ALT (IU/L)	30.1	29.4	28.7	30.8	0.86	29.4	29.4	29.9	30.2	1.21
AST (IU/L)	117	121	114	124	8.0	109	113	125	127	11.3
Goat						109 ^b^	94.3 ^b^	151 ^a^	121 ^ab^	
Sheep						118 ^ab^	132 ^ab^	99.4 ^b^	135 ^ab^	

^1^ PCV, BUN, ALP, ALT, and AST = packed cell volume, blood urea nitrogen, alkaline phosphatase, alanine aminotransferase, and aspartate aminotransferase, respectively; ^2^ Gro = growing, Mat = mature; ^3^ FRW, SW-L, SW-M, and SW-H = fresh water and saline water with 10, 13.5, and 17 g of TDSs/L, respectively; ^4^ SEM = standard error of the mean for species or age. ^a,b^ means between columns without a common superscript letter differ (*p* < 0.05).

**Table 8 animals-14-01565-t008:** Effects of drinking water treatment (WT) with different salinity levels on thermoregulatory traits of growing and mature Blackhead Ogaden sheep and Somali goats.

Item	Species		Age ^1^			Water Treatment ^2^				
	Goat	Sheep	Gro	Mat	SEM ^3^	FRW	SW-L	SW-M	SW-H	SEM
Rectal temperature (°C)	37.7 ^b^	38.2 ^a^	38.0	37.9	0.066	37.9	38.0	38.0	37.9	0.09
Respiration rate (breath/min)	22.4	22.5	22.3	22.6	0.27	22.5	22.8	22.3	22.3	0.38
Pulse rate (beats/min)	87.2	86.7	87.3	86.7	0.45	86.9	87.9	86.2	86.8	0.64

^1^ Gro = growing, Mat = mature; ^2^ FRW, SW-L, SW-M, and SW-H = fresh water and saline water with 10, 13.5, and 17 g TDSs/L, respectively; ^3^ SEM = standard error of the mean for species or age. ^a,b^ Means between columns without a common superscript letter differ (*p* < 0.05).

## Data Availability

Data are contained within the article.

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
