# Peer review of "Effects of Salinity Levels of Drinking Water on Water Intake and Loss, Feed Utilization, Body Weight, Thermoregulatory Traits, and Blood Constituents in Growing and Mature Blackhead Ogaden Sheep and Somali Goats"

_animals, 2024, doi:10.3390/ani14111565_

Round 1

Reviewer 1 Report

Comments and Suggestions for Authors

This paper investigated the effects of salinity levels in drinking water on various aspects of performance and health in Blackhead Ogaden sheep and Somali goats. The study included both growing and mature animals, providing a comprehensive understanding of the effects of salinity across different age groups, and the research provides valuable insights into the effects of salinity on various aspects of livestock performance and health, including water intake and loss, feed intake and digestion, body weight, thermoregulatory traits, and blood parameters. The findings contributed to understanding the impact of salinity on livestock and can be useful for livestock producers and policymakers in areas with increasing water salinity. The paper is well-written and organized, thereforeupon the authors' resolution of the following concerns, I believe that this article can be considered for acceptance.

1. Please check all the formatting of the paper, such as the last row in Table 1 and Table 2 that both have obvious mistakes.

2. There are too many tables in the paper. I believe that tables should be converted into figure as much as possible to facilitate readability.

3. The exact meaning of "BW0.75" in the paper should be clarified by the author.

4. The data for Tables 6, 4, and 2 do not provide the error range of the average values, which hinders readers from assessing the credibility of the data.

5. The ALP (IU/L) data in Table 6 for the SW-H group is as high as 1306, while the article states that there were no significant changes in liver enzymes. This is contradictory. The authors need to carefully check if the data in the table is consistent with the description in the article.

6. The author's statement in the Conclusions section, "Thus, these results suggest that Blackhead Ogaden sheep and Somali goats, which were adapted to arid conditions, may maintain normal physiology by consuming drinking water with up to 17 g TDS/L without adverse effects," is too absolute and lacks rigor.

Author Response

This paper investigated the effects of salinity levels in drinking water on various aspects of performance and health in Blackhead Ogaden sheep and Somali goats. The study included both growing and mature animals, providing a comprehensive understanding of the effects of salinity across different age groups, and the research provides valuable insights into the effects of salinity on various aspects of livestock performance and health, including water intake and loss, feed intake and digestion, body weight, thermoregulatory traits, and blood parameters. The findings contributed to understanding the impact of salinity on livestock and can be useful for livestock producers and policymakers in areas with increasing water salinity. The paper is well-written and organized, thereforeupon the authors' resolution of the following concerns, I believe that this article can be considered for acceptance.

Response: Thanks for positive comments. Drinking water salinity might be a burning problem in the context of climate crisis.

  1. Please check all the formatting of the paper, such as the last row in Table 1 and Table 2 that both have obvious mistakes.

Response: Thank you for your suggestion. The formats in all tables have been uniformed.

  1. There are too many tables in the paper. I believe that tables should be converted into figure as much as possible to facilitate readability.

Response: In our opinion, the quantitative mean data are better to present in tables, which will be useful for readers to understand the magnitude of change or for meta-analysis by someone else. Temporal data are good to present in figures to get trend over time or complex data (e.g., networks, community alterations, etc) to get visualize properly. We hope the reviewer understands our views.

  1. The exact meaning of "BW0.75" in the paper should be clarified by the author.

Response: good suggestions. We have defined it in the manuscript.

  1. The data for Tables 6, 4, and 2 do not provide the error range of the average values, which hinders readers from assessing the credibility of the data.

Response: It is common to use SEM as a pooled standard error of mean, and we presented it in the tables.

  1. The ALP (IU/L) data in Table 6 for the SW-H group is as high as 1306, while the article states that there were no significant changes in liver enzymes. This is contradictory. The authors need to carefully check if the data in the table is consistent with the description in the article.

Response: Appreciated to find out this error. It was a typo in our revised version, not sure how it happened. But in our first version, it was 130. So, it would be 130 instead of 1306.

  1. The author's statement in the Conclusions section, "Thus, these results suggest that Blackhead Ogaden sheep and Somali goats, which were adapted to arid conditions, may maintain normal physiology by consuming drinking water with up to 17 g TDS/L without adverse effects," is too absolute and lacks rigor.

Response: We have revised the conclusion. Hope it is better now.

Reviewer 2 Report

Comments and Suggestions for Authors

The authors examined the effects of drinking water salinity levels on water intake and loss, feed utilization, body weight, thermoregulatory traits, and blood components in growing and mature Blackhead Ogaden sheep and Somali goats. The subject of this study is suitable for the “Animals” journal. The authors indicated that salinity level of drinking water up to 17 g TDS/L of NaCl increases water intake and water loss and reduces nutrient digestibility compared to freshwater but has no effect on feed intake, ADG, thermoregulation, or most blood characteristics. Additionally, the authors suggest that Blackhead Ogaden sheep and Somali goats adapted to arid conditions can maintain normal physiology without any adverse effects by consuming salinity level of drinking water up to 17 g TDS/L of drinking water. The study is well designed and presented, but I offer a few corrections to increase the scientific value of the manuscript. After the authors address these corrections, the manuscript can be accepted.

- Differences exist in the body weight between growing and mature Blackhead Ogaden sheep and growing and mature Somali goat. Did the authors examine these differences statistically? If there is a difference, it should be stated in the results section. Additionally, did the authors consider the possible effects of body weight on other investigated parameters as co-factors in the statistical analysis?

-The number of animals used in the study is stated in the manuscript. However, in the statistical analysis section, a power analysis should show that the number of animals is sufficient. Otherwise, the reliability of the results obtained may be questioned. I would also recommend that the distribution of animals in treatment groups be presented as a table to make it more understandable.

- Reference should be cited for all measurement parameters.

- I request the authors to include information about sensitivity, control concentration used, and coefficient of variation (which evaluates assay precision) in hormonal analyses. How many tests were carried out per hormone? If they were more than 1, in addition to reporting the intra-assay coefficient of variation, it is also essential to report the inter-assay coefficient of variation. This information is crucial to evaluate the test carried out.

-The results have been well presented, and the discussion has supported the results, but the conclusion section should be improved, especially the effect of salinity level of drinking water on animal health, welfare, and yield.

Also;

-The results are convincing and supported by the discussion.

-The conclusions are consistent with the evidence and arguments.

-The topic and references are appropriate.

-Tables and figures are presented clearly and understandably.

Author Response

The authors examined the effects of drinking water salinity levels on water intake and loss, feed utilization, body weight, thermoregulatory traits, and blood components in growing and mature Blackhead Ogaden sheep and Somali goats. The subject of this study is suitable for the “Animals” journal. The authors indicated that salinity level of drinking water up to 17 g TDS/L of NaCl increases water intake and water loss and reduces nutrient digestibility compared to freshwater but has no effect on feed intake, ADG, thermoregulation, or most blood characteristics. Additionally, the authors suggest that Blackhead Ogaden sheep and Somali goats adapted to arid conditions can maintain normal physiology without any adverse effects by consuming salinity level of drinking water up to 17 g TDS/L of drinking water. The study is well designed and presented, but I offer a few corrections to increase the scientific value of the manuscript. After the authors address these corrections, the manuscript can be accepted.

Response: Thanks for your positive evaluation.

- Differences exist in the body weight between growing and mature Blackhead Ogaden sheep and growing and mature Somali goat. Did the authors examine these differences statistically? If there is a difference, it should be stated in the results section. Additionally, did the authors consider the possible effects of body weight on other investigated parameters as co-factors in the statistical analysis?

Response: Yes, initial and final BW, and average daily gain were presented in Table 4. ADG that reflects the BW difference per day did not show significant difference in this study. There was no significant (P = 0.97) difference for the initial BW among the treatment. Therefore, it is not valid to use BW as a co-variance in the statistical analysis. Nonetheless, some variables that may be influenced by BW such as feed intake and water consumption were expressed relative to BW (Tables 2 and 4).

-The number of animals used in the study is stated in the manuscript. However, in the statistical analysis section, a power analysis should show that the number of animals is sufficient. Otherwise, the reliability of the results obtained may be questioned. I would also recommend that the distribution of animals in treatment groups be presented as a table to make it more understandable.

Response: A table has been presented showing the distribution of animals in treatments. We feel 10 animals per treatment (n = 10) within an age and a species is sufficient for this type of study. Many studies with replicates of lower 10 animals have been published. We indeed perform power analysis in our many saline water studies and found 6 to 8 animals per treatment is required with alpha of 0.05 and power of 0.80.

- Reference should be cited for all measurement parameters.

Response: There were references cited for feed and water composition analysis. We revised the citations for blood analysis.

- I request the authors to include information about sensitivity, control concentration used, and coefficient of variation (which evaluates assay precision) in hormonal analyses. How many tests were carried out per hormone? If they were more than 1, in addition to reporting the intra-assay coefficient of variation, it is also essential to report the inter-assay coefficient of variation. This information is crucial to evaluate the test carried out.

Response: In this study, there was no hormonal analysis. Some blood biochemical analyses were performed, which are commonly reported. There were 80 replicates (animals) for each variable. Those blood biochemical variables were within normal physiological ranges.

-The results have been well presented, and the discussion has supported the results, but the conclusion section should be improved, especially the effect of salinity level of drinking water on animal health, welfare, and yield.

Response: Thanks for your positive evaluation. We have revised the conclusion as per your suggestion.

Also;

-The results are convincing and supported by the discussion.

-The conclusions are consistent with the evidence and arguments.

-The topic and references are appropriate.

 -Tables and figures are presented clearly and understandably.

Response: Thanks.